# CRISPR screens in 3D tumourspheres identified miR-4787-3p as a transcriptional start site miRNA essential for breast tumour-initiating cell growth
Tom Stiff[1], Salih Bayraktar[1], Paola Dama [1], Justin Stebbing [2] & Leandro Castellano [1,3] ✉

Our study employs pooled CRISPR screens, integrating 2D and 3D culture models, to identify miRNAs critical in Breast Cancer (BC) tumoursphere formation. These screens combine with RNA-seq experiments allowing identification of miRNA signatures and targets essential for tumoursphere growth. miR-4787-3p exhibits significant up-regulation in BC, particularly in basal-like BCs, suggesting its association with aggressive disease. Surprisingly, despite its location within the 5'UTR of a protein coding gene, which defines DROSHA-independent transcription start site (TSS)-miRNAs, we find it dependant on both DROSHA and DICER1 for maturation. Inhibition of miR-4787-3p hinders tumoursphere formation, highlighting its potential as a therapeutic target in BC. Our study proposes elevated miR-4787-3p expression as a potential prognostic biomarker for adverse outcomes in BC. We find that protein-coding genes positively selected in the CRISPR screens are enriched of miR-4787-3p targets. Of these targets, we select ARHGAP17, FOXO3A, and PDCD4 as known tumour suppressors in cancer and experimentally validate the interaction of miR-4787-3p with their 3'UTRs. Our work illuminates the molecular mechanisms underpinning miR-4787-3p's oncogenic role in BC. These findings advocate for clinical investigations targeting miR-4787-3p and underscore its prognostic significance, offering promising avenues for tailored therapeutic interventions and prognostic assessments in BC.

Breast cancer (BC) is a heterogeneous malignancy, standing as the most prevalent form of cancer among women[1]. Around 70-80% of early-diagnosed BC cases can be treated successfully. However, advanced-metastatic BC remains challenging, currently lacking a cure[1].

Through analysis of gene expression signatures, BC has undergone systematic classification into well-defined molecular subtypes, namely Luminal A, Luminal B [both distinguished by the presence of the Estrogen receptor (ER)], HER2-enriched, and Basal-like. Each of these subtypes exhibits distinct molecular features, clinical characteristics, and therapeutic implications[2].

Furthermore, an alternative classification method is based on immunohistochemistry profiles, wherein BC can be categorised according to the expression of specific biomarkers such as ER, progesterone receptor (PR), and HER2. In cases where ER, PR, and HER2 are not detected, the subtype is referred to as triple-negative BC (TNBC).

Like many other malignancies, BC is characterised by a heterogeneous population of cancer cells. Within the tumour mass, a minor subset comprising only a small fraction (0.1-1%) consists of tumour-initiating cells (TICs), which are also referred to as cancer stem cells (CSCs)[3]. TICs possess the unique ability to undergo self-renewal, leading to the generation of non-tumourigenic, proliferating progeny, as well as the capacity to initiate the formation of new tumours[3]. CSCs are also called TICs, because when inoculated into severe combined immunodeficiency disease (SCID) mice, represent the minority of breast cancer cells capable of forming new tumours and are CD44$^+$/CD24$^{-/low}$/lineage$^-$ [4]. TICs can be isolated or enriched through various methods. This includes sorting breast cancer cells based on the CD44$^+$/CD24$^{-/low}$ phenotype, or culturing cells under non-adherent, non-differentiating conditions to promote the formation of tumourspheres[5]. Interestingly, studies have demonstrated that adding B27

[1]University of Sussex, School of life Sciences, John Maynard Smith Building, Falmer, Brighton, BN1 9QG, UK. [2]Department of Life Sciences, ARU, Cambridge, UK. [3]Department of Surgery and Cancer, Division of Cancer, Imperial College London, Imperial Centre for Translational and Experimental Medicine (ICTEM), London, W12 0NN, UK. ✉e-mail: l.castellano@sussex.ac.uk

to the culture media in non-adherent conditions improves tumoursphere formation efficiency and enhances the enrichment of the CD44 + /CD24–/ low lineage, which can reach up to approximately 95% in breast cancer cell lines. This indicates that using tumoursphere 3D cultures is sufficient to study this tumourigenic cancer cell population.

Given their pivotal role, TICs are considered the driving force behind cancer progression and are implicated in conferring resistance to various therapeutic approaches. Consequently, strategies focused on targeting TICs hold significant promise in combating metastatic cancers and preventing disease recurrence[6]. TICs can be studied by exploiting models based on three-dimensional (3D) culturing conditions and low plating densities to form clonal cultures of TIC-enriched tumourspheres[7,8].

MicroRNAs (miRNAs) are short, non-coding RNA molecules, approximately 22 nucleotides in length, that play a pivotal role in the post-transcriptional control of gene expression[9]. Their biogenesis initiates with the transcription of a primary miRNA (pri-miRNA). In the nucleus, the Microprocessor complex, composed of DROSHA and DGCR8, processes the pri-miRNA into a precursor miRNA (pre-miRNA). Subsequently, the pre-miRNA is cleaved by DICER1 in the cytoplasm to form the mature miRNA. Some miRNAs, however, can bypass the need for the Microprocessor or DICER1 in their biogenesis mechanism[10].

They have emerged as key contributors to various human diseases, with significant implications for different aspects of cancer progression[11]. Through base-pair complementary interactions, they suppress the post-transcriptional expression of mRNA targets. However, each miRNA is predicted to target thousands of mRNAs[12]. Given this extensive capacity for direct gene co-regulation, identifying which of these targets serves as the key effector impacting the phenotype controlled by the miRNAs becomes challenging.

The exploitation of miRNAs as therapeutic targets in tumours holds great promise, owing to all miRNAs being inherently targetable. Oncogenic miRNAs can be effectively and selectively suppressed using synthetic, complementary anti-miRNAs (anti-miRs), which interfere with their function, thus attenuating their detrimental effects on tumourigenesis[11]. Conversely, in the case of tumour suppressor miRNAs, their expression can be enhanced or re-established in tumour cells to restrain tumour growth and potentially reverse cancer-associated phenotypes[11].

The role of miRNAs in cancer, including BC, has primarily been elucidated through miRNA expression profiling studies conducted in cancerous versus normal samples, comparison of tumour stages or within cancer cell lines[13]. Nevertheless, this approach relies on the averaged expression levels of miRNAs, and thus, it may not fully uncover their functional significance. For example, a subset of miRNAs might have the ability to impact cancer vulnerabilities, but their expression remains unchanged between cancer and normal samples. In such cases, experimental conditions used to

discover important miRNAs are based solely on differences in expression levels, such as RT-qPCR, microarray o RNA-seq might overlook important miRNAs.

CRISPR pooled screens have allowed for uncovering pivotal cancer gene drivers, revolutionising our comprehension of the intricate mechanisms through which genes and pathways contribute to tumourigenesis[14]. However, CRISPR screens have predominantly been carried out in 2D cell cultures, which often fail to fully capture the intricate complexities of tumour biology[15]. To help address these limitations, recent studies have conducted screens in 3D cultures[15,16]. These investigations have demonstrated that screens exhibiting stronger effects in 3D cultures, as compared to 2D cultures, are enriched for genes that have been found to be mutated in cancer[15]. This finding emphasises the potential of CRISPR screens in 3D cultures as an ideal model for effectively identifying key players in cancer.

Importantly, despite their transformative impact, to date, CRISPR screens have not been exploited for identifying the miRNAs that exert influence on breast tumourigenesis. miRNA-focused CRISPR screens may unveil novel therapeutic targets and pivotal molecular pathways[11].

To discover the miRNA-dependent vulnerabilities in breast tumourigenesis, we conducted genome-wide CRISPR-CAS9 screens targeting genes and miRNAs in both ER+ and TNBC cancer cells, employing both 2D monolayer and 3D tumoursphere models (Fig. 1a). This approach allowed us to reveal the miRNAs and their critical targets controlling TIC viability in BC. By therapeutically addressing these critical miRNAs, we hold the potential to combat metastatic cancers and significantly reduce the risk of disease recurrence[6].

## Results

### CRISPR-CAS9 screens in 3D tumourspheres versus 2D cultures identify miRNA-dependent vulnerabilities of breast TICs

Given that 3D tumourspheres are enriched with TICs, which are able to self-renew, and are known to play a pivotal role in tumour progression[3], we conducted simultaneous CRISPR-Cas9 screens in two BC cell lines under both 3D tumoursphere and 2D growth conditions to identify the miRNAs and their targets that affect tumoursphere growth. (Fig. 1a). By using both culture types in the screens we could specifically identify factors that influence 3D breast tumoursphere growth. To ensure a comprehensive exploration across two BC subtypes, we utilised MCF7 cells, representing the luminal A subtype (ER+, PR+), and HCC1395 cells, which belong to the aggressive TNBC subtype (ER-, PR-, HER2-). For the screens, we used the Genome-Scale CRISPR Knockout (GeCKO) v2 library[17]. This library allows us to systematically target a total of 19,052 genes and 1,864 miRNAs[17], enabling us to identify the miRNAs and their crucial target transcripts essential for breast tumourigenesis.

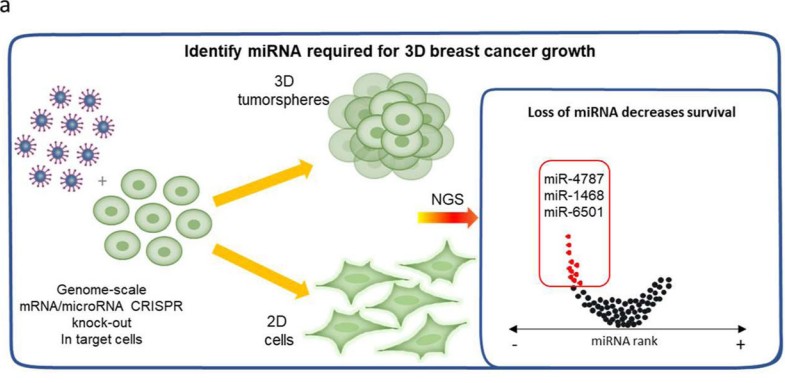

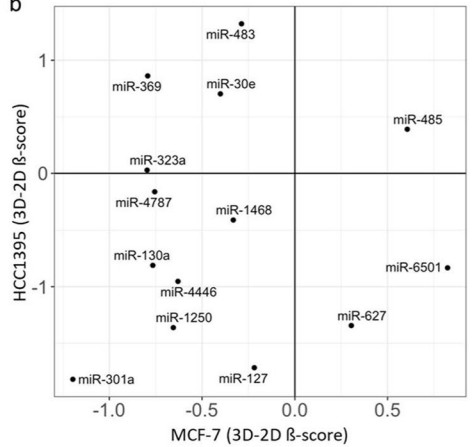

**Fig. 1 | CRISPR screening strategy and performance. a** Schematic of the genome-wide screening strategy in the 2D and 3D tumoursphere culture. **b:** Correlation of miRNA 3D-2D Gecko screen MAGeCK Beta scores between the cell lines MCF7 and HCC1395.

To perform this experiment, we first optimised 3D tumourspheres for large-scale culture necessary to conduct the screens. Next, we subjected the cells to lentiviral infection with the pooled GeCKO v2 library (MOI 0.3) and applied antibiotic selection, then split the screen into 2D/3D growth conditions and cultured for 4 weeks. After this step, we conducted next-generation sequencing (NGS) to analyse the incorporation of single guide RNAs (sgRNAs) into the genome. Subsequently, we utilised Model-Based Analysis of Genome Wide CRISPR-CAS9 Knockout (MAGeCK)[18] to analyse the NGS data. The screen performance was excellent with over 80% of the library mapped at time 0 (T0) and in the 2D or 3D samples that we left to grow for 4 weeks, in both cell lines. For ER + MCF-7 cells, using MAGeCK analysis we identified negatively selected sgRNAs (FDR < 0.05 in 3D condition versus T0) targeting 235 protein coding genes (PCGs) with a negative 3D-2D ß-score cutoff of < -0.4 and positively selected sgRNAs that target 17 PCGs with a positive 3D-2D ß-score of > 0.4 (Supplementary Data 1). Furthermore, for TNBC, HCC1395 cells, we identified negatively selected sgRNAs targeting 320 PCGs (FDR < 0.05 in 3D condition versus T0) with a negative 3D-2D ß-score < −0.4 and positively selected sgRNAs targeting 24 PCGs with a positive 3D-2D ß-score > 0.4 (Supplementary Data 1). Supporting the validity of our CRISPR screens, the identified PCGs with 3D-2D ß-score < -0.4 are known to be important for TIC expansion and stemness (Fig. S1). Accordingly that, pathway enrichment analysis using EnrichR[19] showed that in both MCF-7 and HCC1395, genes with negative 3D-2D ß-score were enriched for genes that have MYC interacting with their promoter in both mouse embryonic stem cells (mESCs) or BC cells (Fig. S1a, d). Correspondingly, it is known that MYC controls a transcriptional program promoting self-renewal in both mESCs or BC cells[20,21]. Furthermore, genes with negative 3D-2D ß-score in MCF-7 where mostly enriched for regulation of cholesterol biosynthesis pathways, which has been shown to promote TICs expansion and tumour progression in BC[22,23] (Fig. S1b, c). Interestingly, in HCC1395 there was a specific enrichment in PCGs involved in telomere maintenance and extension (Fig. S1g).

### The screen identified selected miRNAs which KO produces large negative ß-scores in BC 3D cultures

Next, to capture essential miRNAs for 3D growth, we considered the miRNAs that have large and significant negative ß-scores in 3D culture in both MCF-7 and HCC1395 cells (Table S1). We selected the top 7 miRNAs per cell line that have the lowest negative ß-scores in 3D culture for the specific cell line, are successfully KO by more than 3 gRNAs after infection with the Gecko library and are expressed in the cell lines and in BC specimens. To evaluate miRNA expression in tumour specimens and cell lines, we used miRNA-seq data from the DIANA-miTED[24], the TCGA and the NCBI GEO databases (GSE100769)[25].

Those we observed with large negative ß-scores are acting as onco-miRs, meaning that in the 3D culture conditions they are normally acting to drive 3D growth. We decided to focus on this group as the endogenous miRNAs could be potentially and selectively targeted for therapeutic intervention. Most of those miRNAs (miR-4787, miR-1468, miR-130a, miR-4446, miR-1250, miR-127 and miR-301a) have negative 3D-2D ß-scores in both cell lines (Fig. 1b), indicating that they specifically affect tumoursphere growth and that they can be potentially used to target both ER+ and TNBC types. We also identified miRNAs which are specifically essential for MCF-7 3D tumoursphere growth (miR-323a, miR-369, miR-30e and miR-483) or only for HCC1395 3D tumoursphere growth (miR-6501 and miR-627) (Fig. 1b).

### Validation of miRNA screen hits using inhibitors and measuring 3D versus 2D growth

In order to validate the hits, we tested the effect of miRNA inhibition on cells grown in monolayers (2D) and cells grown in 3D spheres for both cell lines. To this end, we first measured the abundance of the miRNAs in BC specimens and cell lines, and we specifically inhibited the most abundant miRNA arm of each miRNAs using ZEN modified 2′-O-methyl RNA steric blocking oligonucleotides to test their phenotypic effect in BC. Cells were

transfected with the specific inhibitors every 4 days for 2 weeks following measurement of 2D or 3D growth (Fig. 2a–d). We observed small changes in the 2D growth phenotype but nothing of statistical significance. In contrast we did observe a significant decrease in 3D tumoursphere growth after targeting some of the miRNA hits (Fig. 2a,b) with representative images of MCF7 tumourspheres shown in Fig. 2e (HCC1395 tumourspheres had very similar gross morphology). MiR-301a-3p, miR-483-3p, miR-1468-5p, miR-4787-3p significantly reduced tumoursphere growth in both MCF-7 and HCC1395. In contrast inhibition of miR-6501-5p and miR-627-5p significantly reduced tumoursphere growth in MCF-7 rather than HCC1395. Whereas inhibition of miR-4446-3p specifically reduced the formation of HCC1395 tumourspheres. Based on these results, we selected miR-301a-3p, miR-483-3p, miR-627-5p, miR-1468-5p, miR-4446-3p, miR-4787-3p and miR-6501-5p for further investigation.

### High expression of these miRNAs negatively correlates with BC patient survival

To evaluate the clinical significance of the selected miRNAs we first examined their expression values in BC in combination with survival clinical data from The Cancer Genome Atlas (TCGA) (https://www.cancer.gov/tcga) and the METABRIC (https://ega-archive.org/studies/EGAS00000000122) databases. Amongst selected miRNAs, they were all present in the TCGA (Fig. 3a–g), but only miR-301a and miR-627 were measured in the METABRIC study (Fig. S2). Therefore, we could only test these two miRNAs from the METABRIC study. In summary, with the exception of miR-627, whose high expression correlates with a favourable prognosis, the expression levels of the other miRNAs serve as predictors of an unfavourable prognosis, which is in line with their oncogenic role predicted by our CRISPR screen.

### Unsupervised hierarchical clustering of 6 miRNAs from the CRISPR signature is sufficient to separate BC molecular subtypes

Next, we used miRNA-seq data from the BC study from the TCGA database (TCGA-BRCA, https://portal.gdc.cancer.gov/projects/TCGA-BRCA) to evaluate the expression of those miRNAs in tumours versus matched normal samples (Fig. 3h). The expression of miR-301a, miR-4446, miR-4787 and miR-6501 was higher in cancers in comparison to normal samples (p < 0.00001). In contrast, the expression of miR-1468 and miR-483 was lower in cancer compared to normal (p < 0.00001), whereas the expression of miR-627 did not change between the two conditions (Fig. 3h). We then used unsupervised hierarchical clustering, based on the expression of all these miRNAs except from miR-627 [no change between cancer and normal (Fig. 3h) nor a predictor of worse survival (Fig. 3d and Fig. S2b)], to assess whether the expression of these 6 miRNAs in BC (n = 1149) was sufficient to detect the different BC molecular subtypes. We identified that they were able to separate Basal-like subtypes from Normal samples and ER + BC but could not distinguish HER2-enriched tumours (Fig. 3i). Interestingly, this miRNA signature identified a subgroup of Luminal A which clusters with Normal samples that could be classified as a distinct BC subtype. Next, we assessed the differential expression of these miRNAs in BC molecular subtypes. This analysis identified that the expression of those miRNAs in different molecular subtypes is variable and found that only miR-4787, miR-301a and miR-6501 are highly expressed in TIC-enriched Basal-like cancers (Fig. S3).

### RNA-seq analysis on cells treated with miRNA inhibitors reveals repression of cell cycle and growth-related genes as a general mechanism of TICs expansion controlled by essential miRNAs

To investigate the mechanism of action of these miRNAs in BC, we subjected MCF-7 and HCC3519 tumourspheres to miRNA inhibitors targeting the most abundant miRNA arm (anti-miR-301a-3p, anti-miR-4787-3p, miR-1468-5p, miR-4446-3p, miR-483-3p and miR-6501-5p) or a fixed negative control sequence inhibitor (NC1 from IDT) in each cell line, followed by RNA-seq experiments. Using DESeq2, we identified differential

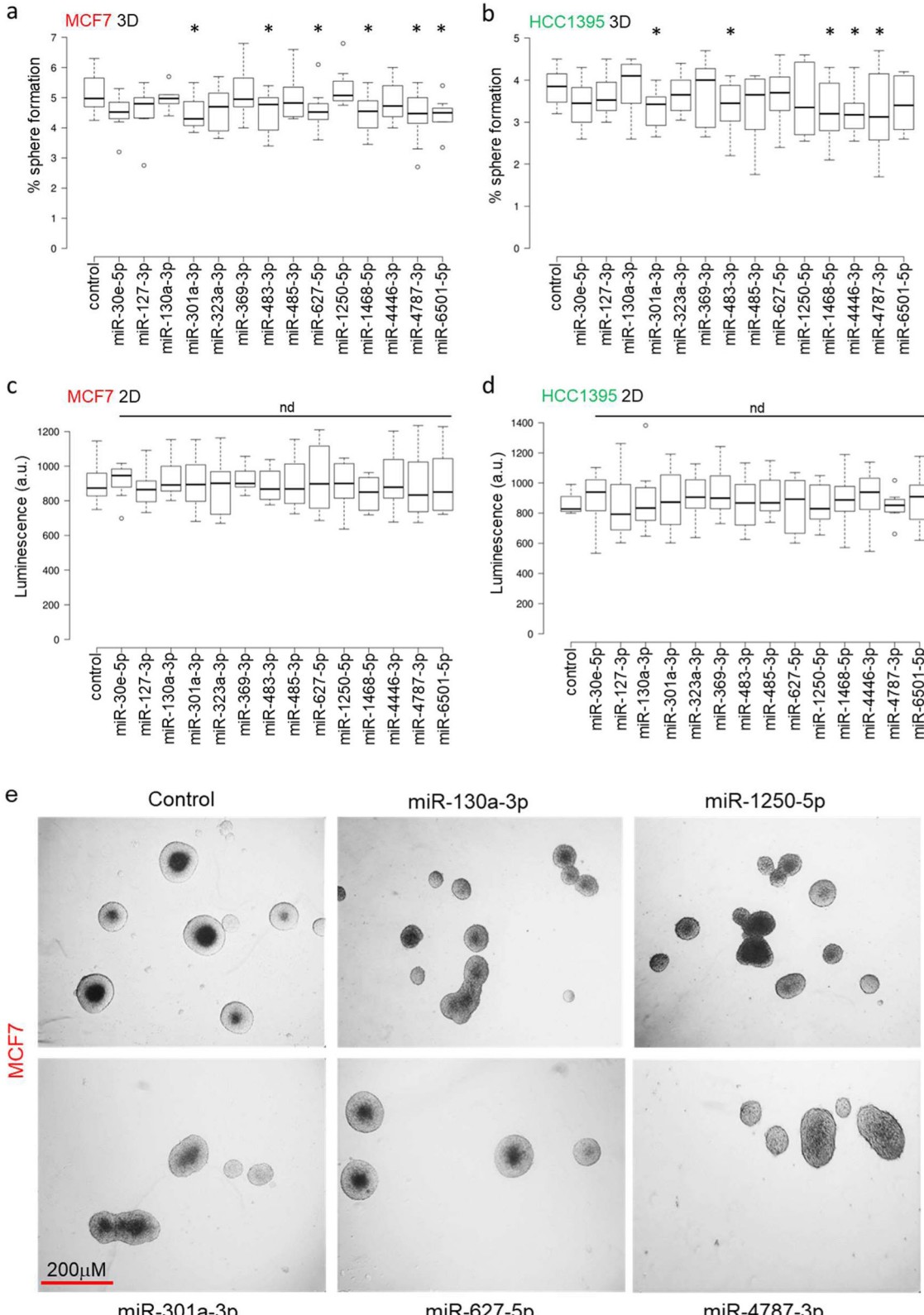

**Fig. 2 | 2D and 3D growth perturbation with miRNA inhibition. a–d**: Growth of MCF7 (red) and HCC1395 (green) in 2D and 3D conditions after inhibition of the indicated miRNAs. For each condition 3 independent biological replicates were performed. **e** Representative images of tumourspheres formed in MCF7 in the indicated conditions. Box plot: Center lines show the medians; box limits indicate the 25th and 75th percentiles as determined by R software; whiskers extend 1.5 times the interquartile range from the 25th and 75th percentiles, outliers are represented by dots.

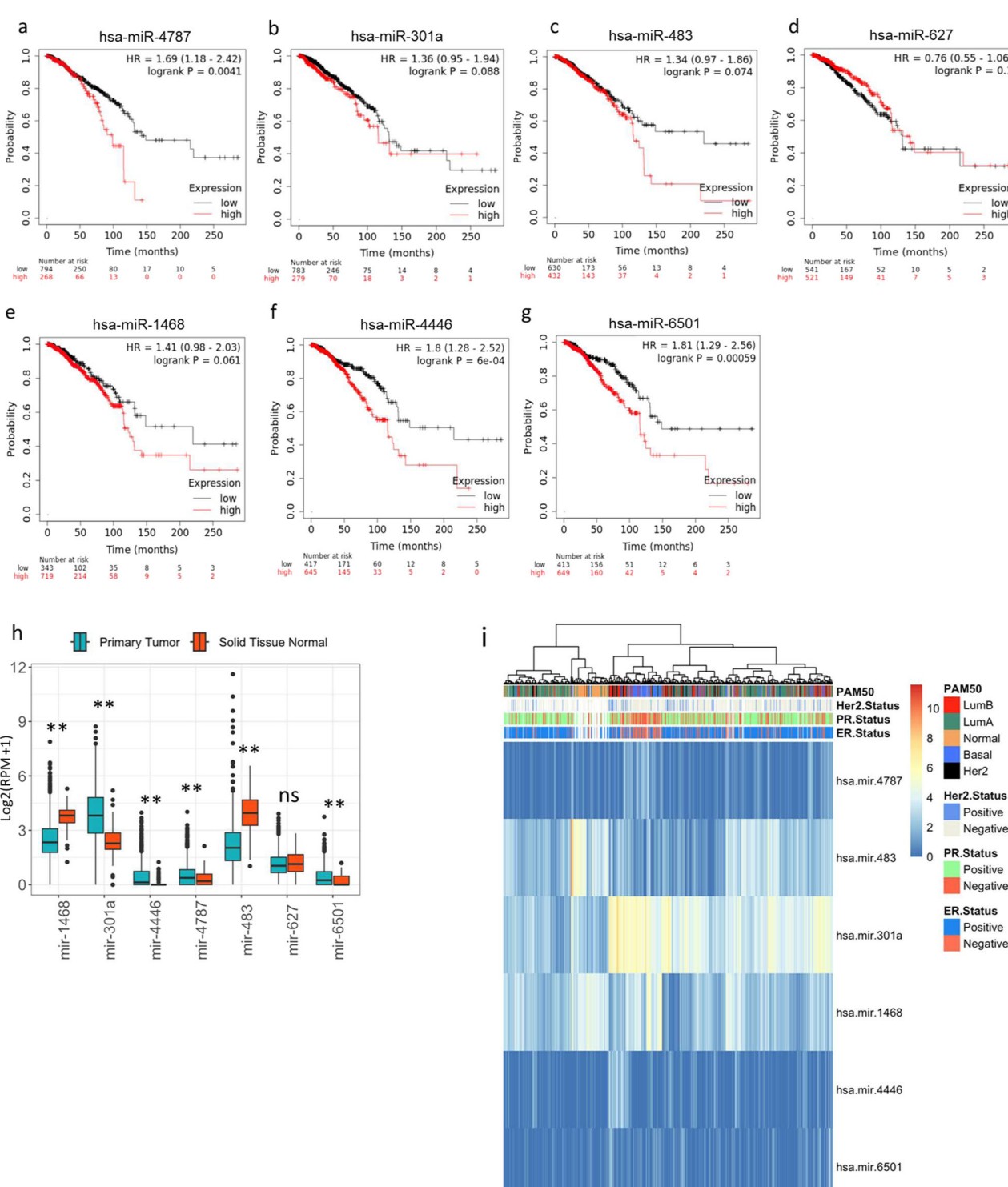

**Fig. 3 | miRNA involvement in breast cancer. a–g** Expression values of miRNA precursors in BC in combination with survival clinical data from The Cancer Genome Atlas (TCGA). **h** Data from the TCGA-BRCA pre-miRNA expression profiles to evaluate the expression of miRNAs in tumours versus matched normal samples. **i** Unsupervised hierarchical clustering based on miRNA expression.

gene expression resulting from the inhibition of each of these miRNAs in tumourspheres from both BC subtypes (Supplementary Data 2). Subsequent GSEA enrichment analysis unveiled the molecular pathways regulated by these miRNAs in both ERα+ and TNBC cell lines (Fig. S4). The genes upregulated upon inhibition of these miRNAs were found to be associated with cell cycle activation and growth, specifically involving Myc targets, E2F targets, and oxidative phosphorylation (Fig. S4). These findings suggest a common mechanism wherein these miRNAs repress highly proliferating cancer cells, thereby promoting the growth of low-proliferating TICs[8].

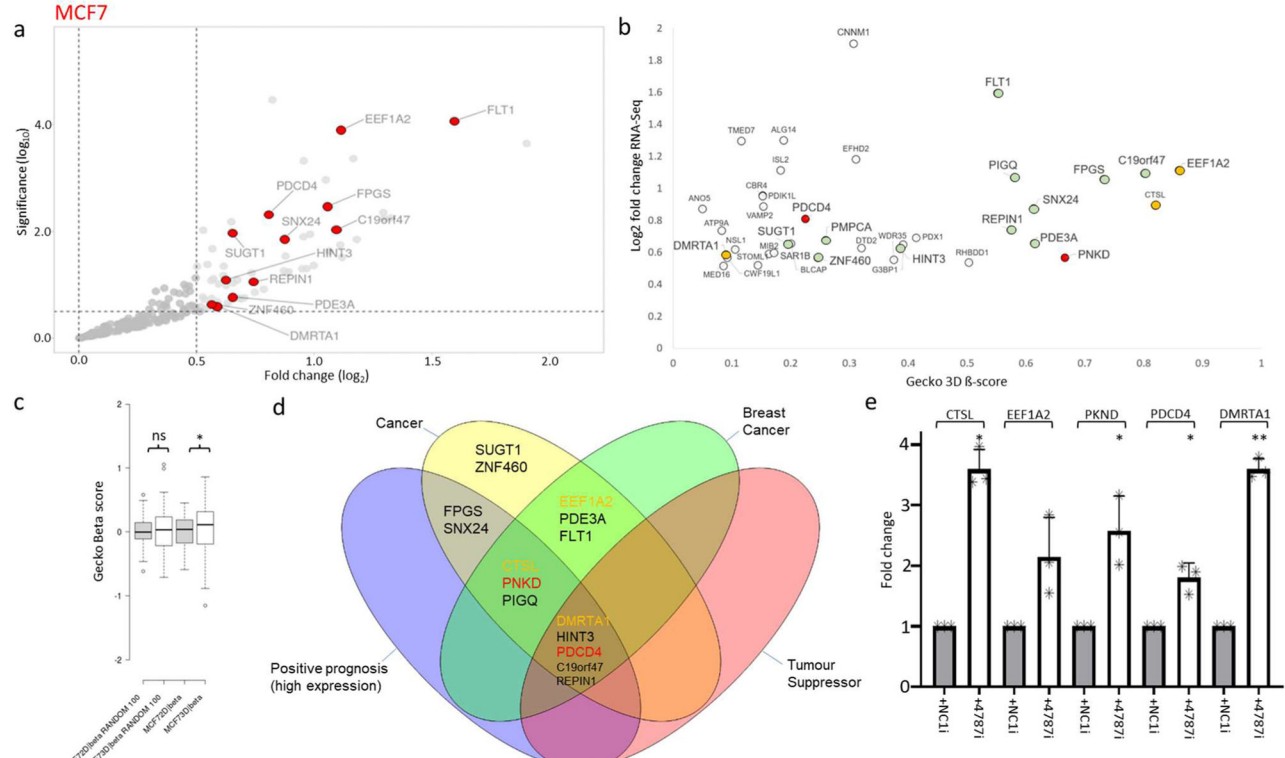

**Fig. 4 | Gecko screen data combines with RNA-seq after miRNA inhibition (MCF7).** **a–b** RNA-seq data after miR-4787-3p inhibition in MCF7 cells (red frame), combined with GeCKO screen NGS data to discover significant hits. **c** Mir-4787 targets have significantly higher Gecko Beta scores in 3D vs 2D culture conditions. **d** Top hits from the combined screens data are highly enriched in relevant cancer related categories. **e** qPCR analysis of MCF7 top 5 targets. Fold increase in expression between negative control inhibitor (NC1i) and miR-4787-3p inhibitor (4787i). Genes of interest in (**b**) are coloured in relation to the analysis in (**d**). Green hits are linked to cancer. Orange hits were validated in (**e**). Red hits were validated in (**e**) and further investigated.

## miR-4787-3p is a DROSHA and DICER1-dependent TSS-miRNA that induces tumourigenesis by targeting genes associated with a positive 3D-2D β-score in CRISPR screens, including important tumour suppressors

We next focused on miR-4787-3p, as its role in BC had not been previously described. Interestingly, miR-4787-3p maps closely and downstream of the transcriptional start site (TSS) of the protein-coding gene DOCK3, (Fig. S5a). Therefore, it can be considered a TSS-miRNA[26,27]. This class of miRNAs has been previously described[26,27], but never associated with cancer, prompting us to further characterise its role in this pathogenesis.

To this end, we conducted comprehensive miR-4787-3p target identification by data integration from the RNA-seq and the CRISPR screens that we performed in BC cell lines following validation studies. In line with the fact that miR-4787-3p originates from the DOCK3 transcript, the expression of miR-4787-3p and DOCK3 shows a positive correlation in BRCA specimens from the TCGA (Fig. S5b; $r = 0.45$, $p < 2.2e−16$). The biogenesis of miRNAs that map within the 5'UTR of protein-coding genes has been described to be independent from DROSHA and therefore from Microprocessor complex, but dependent on DICER1[26,27]. miR-4787-3p is expressed in MCF-7 or HCC3519 BC cell lines, but also in HCT116 and A549 that are colon and lung cancer cell lines respectively (Fig. S5c). To evaluate whether similarly to other TSS-miRNA the biogenesis of miR-4787-3p is Microprocessor independent but DICER1 dependent, we measured its expression in HCT116 cells that have been knockout (KO) for DROSHA or DICER1 through CRISPR/CAS9 editing (Fig. S5d). However, we observed that miR-4787-3p is not expressed in both DROSHA and DICER1 KO cells, indicating that this particular TSS-miRNA undergoes canonical miRNA maturation.

Next, considering that miRNAs act by repressing gene expression post-transcriptionally[10], we used the RNA-seq experiments performed after inhibition of miR-4787-3p in MCF-7 or HCC1395 BC cells to identify in silico predicted miR-4787-3p targets by TargetScan version 8.0 (v8.0)[12] that are up-regulated by miR-4787-3p silencing. This signature has a high probability of being enriched in direct miR-4787-3p targets. We then cross-examined the behaviour of those targets in the CRISPR screens when grown in 3D tumoursphere culture conditions. Therefore, we filtered the entire RNA-seq dataset from MCF-7 cells for hits that are putative targets of miR-4787-3p and showed a significant positive fold change (FDR < 0.05) after miR-4787-3p silencing (Fig. 4a). Furthermore, to identify the key targets that the miRNA represses for TIC formation through 3D tumourspheres, we selected hits that were also identified in the CRISPR 3D screens and showed a high positive ß-score. From that pool we then ranked the hits by RNA-seq log2FC and significance and CRISPR ß-score in the 3D conditions (Fig. 4b). Interestingly, we observed a small but significant increase in expression for all the miR-4787-3p targets found in the CRISPR screen 3D condition compared to the 2D condition, with no significant increase in expression of a random gene list from the CRISPR screen 2D vs 3D (Fig. 4c). This indicates that miR-4787-3p targets tend to have a positive 3D-2D β-score and further confirming the importance of miR-4787-3p in promoting TIC formation and tumourigenesis.

The combined unbiased selection produced a final list of 18 significant targets. Querying these targets in the scientific literature and using the human protein atlas (proteinatlas.org) we found that a large majority (88%) were associated with cancer, with 61% specifically associated with BC (Fig. 4d). We refined the list to a few top hits that has both BC and tumour suppressor association or that showed a positive prognosis when their expression is high (Fig. 4d). We conducted qPCR validation experiments on

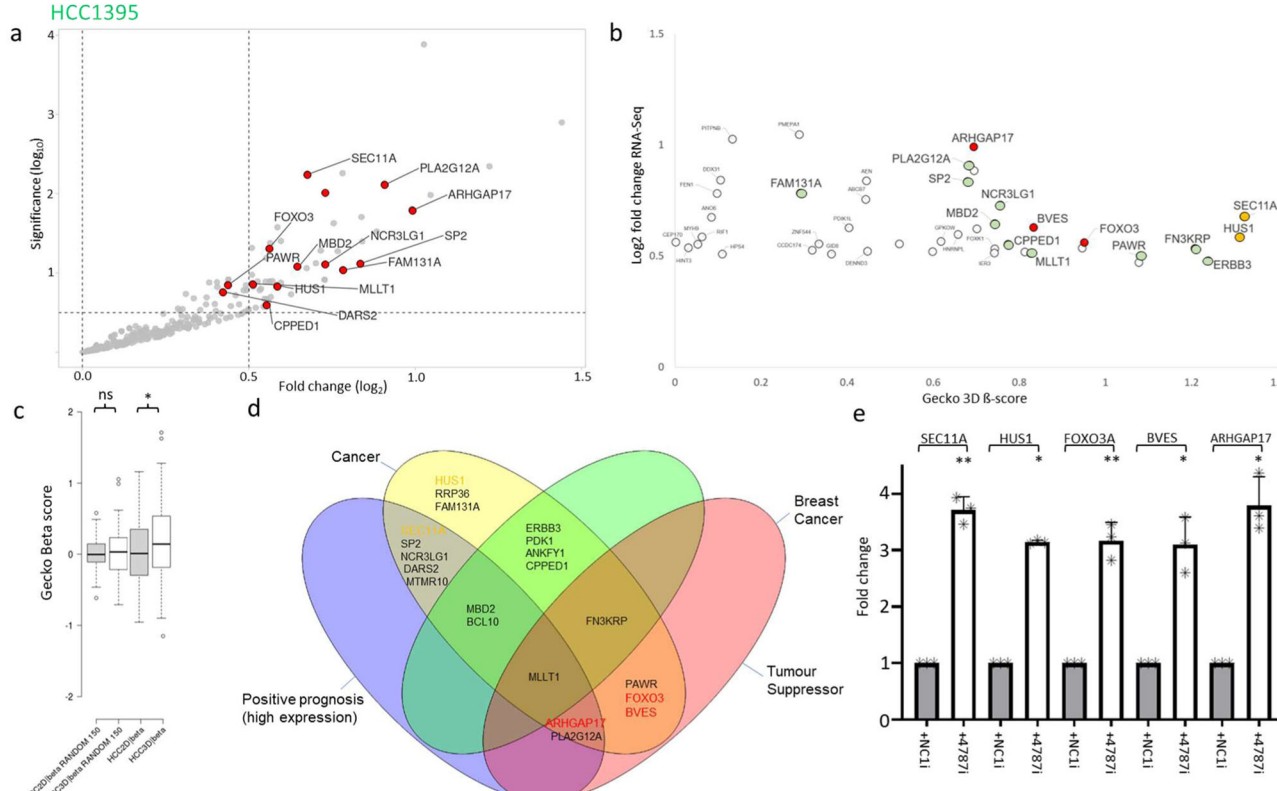

**Fig. 5 | GeCKO screen data combines with RNA-seq after miRNA inhibition (HCC1395). a, b** RNA-seq data after miR-4787-3p inhibition in HCC1395 cells (green frame), combined with GeCKO screen NGS data to discover significant hits. **c** Mir-4787 targets have significantly higher GeCKO Beta scores in 3D vs 2D culture conditions. **d** Top hits from the combined screens data are highly enriched in relevant cancer-related categories. **e** qPCR analysis of HCC1395 top 5 targets. Fold increase in expression between negative control inhibitor (NC1) and miR-4787-3p inhibitor. Genes of interest in (**b**) are coloured in relation to the analysis in (**d**). Green hits are linked to cancer. Orange hits were validated in (**e**). Red hits were validated in (**e**) and further investigated.

the top 5 targets of miR-4787-3p (Fig. 4e). This confirmed the RNA-seq results, showing an increase in target expression after miR-4787-3p inhibition under 3D growth conditions. From those we randomly selected 2 targets for further experimental validation: PKND and PDCD4.

Similarly, we combined the Gecko 3D screen and RNA-seq data for miR-4787-3p in HCC1395 cells. As before, we filtered the entire RNA-seq dataset for hits that are predicted targets of miR-4787-3p and showed a significant positive fold change after miR-4787-3p inhibition (Fig. 5a), then selected those hits that also were found in the GeCKO 3D screen and produced a high positive ß-score (Fig. 5b). We then ranked the hits by RNA-seq log2FC and significance or CRISPR screen ß-score in the 3D or 2D/3D conditions. Consistently with results from MCF-7 (Fig. 4c), we observed a significant increase in all the miR-4787-3p targets found positive in the CRISPR screen 3D condition compared to the 2D condition, with no significant increase in a random list from the CRISPR screen 2D vs 3D (Fig. 5c), reinforcing the importance of miR-4787-3p in promoting TIC formation and tumourigenesis in both ER+ and TNBC cell lines.

The selection process produced a final list of 22 significant targets. Querying these targets in the scientific literature and using the human protein atlas (proteinatlas.org) we found that a large majority (95%) were associated with cancer, with 36% specifically associated with BC (Fig. 5d). We refined the list to top hits that have both cancer and tumour suppressor association (Fig. 5d). We conducted qPCR validation experiments on the top 5 targets of miR-4787-3p (Fig. 5e). This confirmed the RNA-seq results, showing an increase in target expression after miR-4787-3p inhibition under 3D growth conditions. From those we randomly selected 3 targets for further experimental validation: FOXO3A, BVES and ARHGAP17.

In order to test whether the final hits from the screen are indeed direct targets of miR-4787-3p we evaluated whether miR-4787-3p interacts with the target genes within their 3' untranslated regions (3'-UTR) through the miR-4784-3p-mRNA seed complementarity predicted by TargetScan v8.0. To this end we transfected 3'-UTR reporter plasmids of the putative gene targets into MCF7 cells and performed luciferase activity assays (Fig. 6a). When we co-transfected with an RNA mimic for miR-4787-3p we saw a reduction in luciferase activity for all target reporters, suggesting that the miRNA can indeed bind to the target site and inhibit the 3'-UTR. The reduction in activity was significant for: ARHGAP17, FOXO3 and PDCD4. The miR-4787-3p mimic had no significant effect on the luciferase activity of a GAPDH reporter and a random negative control reporter construct (Fig. 6a).

In contrast, a control mimic had no effect on any of the reporter constructs luciferase output, suggesting the effect is specific for miR-4787-3p. The inhibitory effect of the miR-4787-3p mimic could be abolished by co-treating with an inhibitor against miR-4787-3p (Fig. 6a), providing further evidence for a miR-4787-3p specific effect.

As a further control we produced mutated versions of the ARHGAP17, FOXO3A and PDCD4 reporters with the miR-4787-3p seed region mutated. Using these reporters, the miR-4787-3p mimic was now not able to impact on the luciferase signal reported, indicating that it is indeed binding to the miR-4787-3p seed sequence (Fig. 6a).

We sought to further validate these targets of miR-4787-3p. We first overexpressed the identified targets of miR-4787-3p: ARHGAP17, FOXO3A, and PDCD4, in both MCF7 and HCC1395 cell lines (Fig. 6b, c; Fig S6a, b). We observed a significant reduction in 3D tumoursphere formation upon overexpression of each gene in both cell lines, validating our model.

Furthermore, we performed knockdown experiments targeting ARHGAP17, FOXO3A, and PDCD4 in combination with miR-4787-3p inhibition (4787i) in both cell lines (Fig. 6d, e, Fig S6c, d). This allowed us to

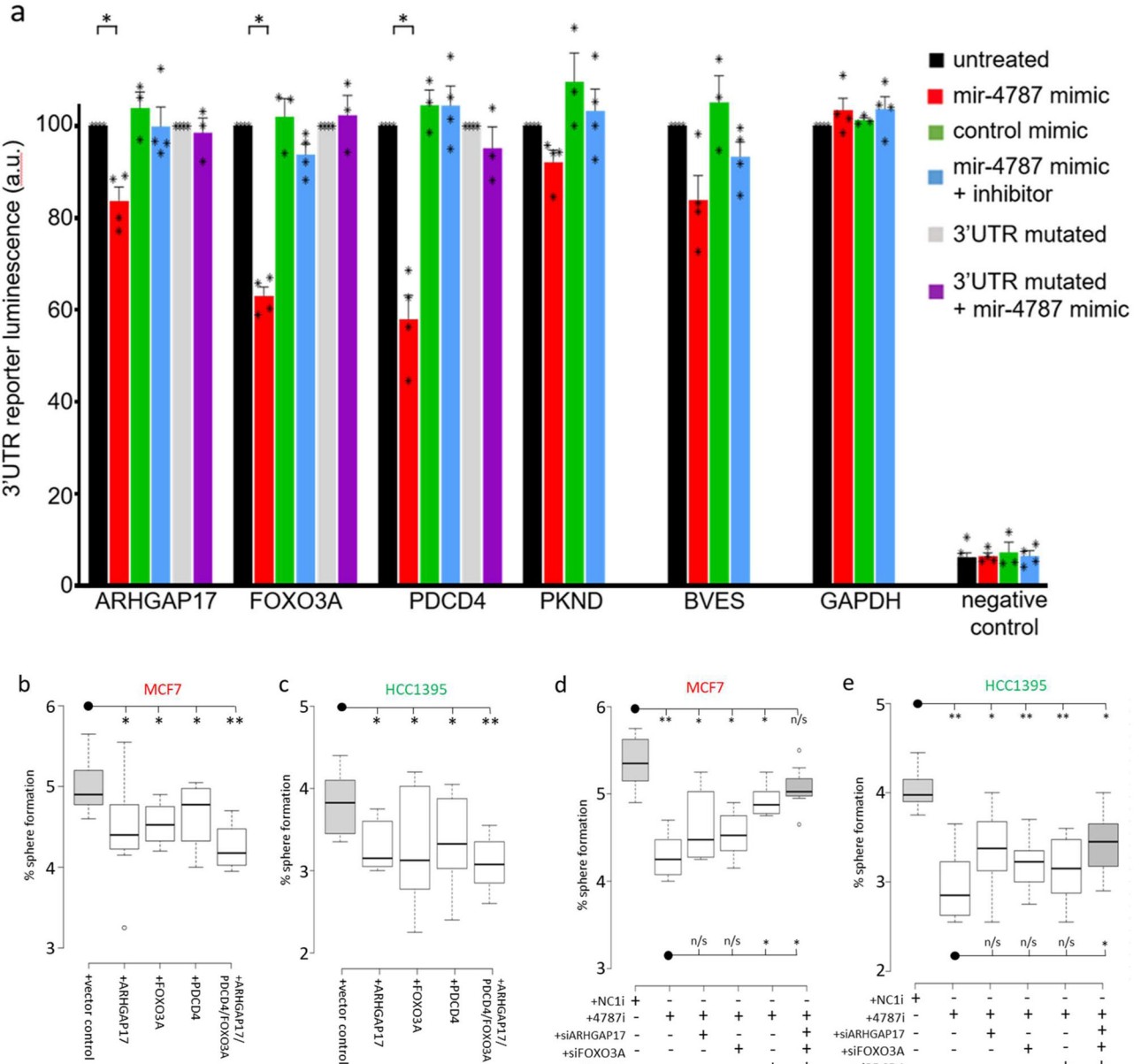

**Fig. 6 | miR-4787-3p targets validate. a** 3'UTR luciferase reporter assay on potential miR-4787-3p targets. Reporter plasmids of putative gene targets were transfected into MCF7 cells and luciferase activity assayed. The reporter was co-transfected with: (**a**) control mimic, miR-4787 mimic, miR-4787 mimic and miR-4787 inhibitor. For ARHGAP17, FOXO3A and PDCD4 a mutated reporter was engineered and also tested for luciferase activity. **b, c** Overexpression of miR-4787-3p targets in MCF7 (**b**) and HCC1395 (**c**) cells reduces tumoursphere formation efficiency. **d–e** Inhibition

of miR-4787-3p reduces tumoursphere formation compared to a negative control inhibitor (NC1) in MCF7 (**d**) and HCC1395 (**e**) cells. Concurrent interference of miR-4787-3p targets using specific siRNA partially rescues the tumoursphere formation defect phenotype. Box plot: Center lines show the medians; box limits indicate the 25th and 75th percentiles as determined by R software; whiskers extend 1.5 times the interquartile range from the 25th and 75th percentiles, outliers are represented by dots.

evaluate the importance of these targets in mediating the 3D tumour formation phenotype induced by miR-4787-3p. Remarkably, inhibition of miR-4787-3p in combination with the repression of these targets significantly rescues the 3D sphere phenotype, particularly when all three targets were co-depleted along with miR-4787-3p inhibition. Interestingly, in MCF7, miR-4787-3p mainly acts through the inhibition of PDCD4, however, when all three targets are inhibited, the sphere formation defect is completely rescued, because it is no longer significant compared to the non-targeting control (NC1i) (Fig. 6d). In contrast, in the triple-negative breast cancer cell lines HCC1396, miR-4787-3p mainly acted through the regulation of ARHGAP17. However, only when all three targets were co-inhibited, the sphere formation defect was partially rescued compared to inhibition of miR-4787-3p alone (Fig. 6e).

## Discussion

In this study, we conducted pooled CRISPR screens employing libraries of sgRNAs targeting a total of 19,052 genes and 1,864 miRNAs using cells grown in monolayers (2D) and cells forming tumourspheres in 3D. To our knowledge, this experimental setup has never been employed to identify miRNAs and their target transcripts crucial in tumourigenesis. We specifically focused on BC, utilising two BC cell lines, MCF7 and HCC1395, representative of two distinct BC subtypes: ER+ and TNBC. To identify essential miRNAs for tumoursphere growth, post the pooled CRISPR screen, we selected miRNAs exhibiting the lowest ß-score in the 3D culture condition. This analysis enabled the identification of a novel miRNA signature crucial for BC tumoursphere growth. In summary, further experimental validation led us to discover that miR-301a-3p, miR-483-3p, miR-

1468-5p, and miR-4787-3p are indispensable for tumoursphere growth in both MCF-7 and HCC1395. Conversely, inhibiting miR-6501-5p and miR-627-5p was crucial for tumoursphere growth in MCF-7 but not in HCC1395. Notably, inhibiting miR-4446-3p specifically reduced the formation of HCC1395 tumourspheres. Interestingly, when integrating expression values of this miRNA signature with unsupervised hierarchical clustering algorithms using the TCGA-BRCA dataset, we successfully differentiated Luminal A, B, and Basal BC subtypes (Fig. 3i). Importantly, we identified miR-301a, miR-4787, miR-4446, and miR-6501 as predictors of poor prognosis (log-rank $p < 0.005$; Fig. 3 and Fig S2), suggesting their potential as prognostic biomarkers in BC. In line with our findings, high expression of miR-301a promotes breast tumourigenesis and predicts poor prognosis[28]. It acts by inducing master regulators of self-renewal, tumourigenesis, and metastasis through the CPEB1/SIRT1/SOX2 axis and downregulating MEOX2[28–30]. The other identified miRNAs have been only marginally described in BC. miR-4446 is part of a miRNA signature predicting high-risk BC cases from liquid biopsies[31], while miR-6501 serves as a diagnostic and prognostic indicator for BC[32]. Finally, miR-1468 has been identified as part of a miRNA signature predicting the stage of breast cancer[33]. Whereas miR-4787-3p, has not previously been associated with the disease. To evaluate the role of miR-301a, miR-4787, miR-4446, miR-6501, miR-1468, and miR-483 in BC, we inhibited the expression of their predominant miRNA arm in BC specimens (miR-301a-3p, miR-483-3p, miR-627-5p, miR-1468-5p, miR-4446-3p, miR-4787-3p, and miR-6501-5p), alongside a scrambled control inhibitor, and conducted RNA-seq. Intriguingly, we observed that genes up-regulated following the inhibition of these miRNAs were predominantly involved in cell growth and proliferation, especially enriched in MYC or E2F targets (Fig S4). As we and others have indicated[8], downregulation of genes crucial for cell growth can lead to the formation of low-proliferating TICs[34]. This suggests that miRNAs induce tumourigenesis by paradoxically repressing genes promoting cell proliferation and growth. Our subsequent focus on miR-4787-3p stems from its never-before-evaluated role in BC. We found significant up-regulation of this miRNA in BC versus healthy control samples (Fig. 3), particularly in basal-like BCs (Fig. S3), the most aggressive subtype and highly enriched in TICs[35]. Interestingly, miR-4787-3p is situated within the 5'UTR region of the protein-coding gene DOCK3. It is known that pre-miRNAs mapping close to the TSS of host genes, like miR-4787-3p, are 7-methylguanosine (m7G) capped and processed by DICER1, but their maturation occurs independently from the Microprocessor complex[26,27]. Contrary to DICER1 processing of canonical pre-miRNAs, DICER1 cleavage of TSS-miRNA generates a single 3p-miRNA. However, our analysis by measuring miR-4787-3p in cells KO for DROSHA and DICER1 unveiled its dependence on both (Fig S5). Additionally, although pre-miR-4787-3p maps within the 5'UTR of DOCK3, it is located 166 nucleotides from its TSS, suggesting that that length include enough nucleotide sequence of the 5'UTR to become a Microprocessor complex substrate. We then integrated RNA-seq data upon miR-4787-3p inhibition with the CRISPR screens conducted in both MCF-7 and HCC1395, along with a miRNA-target identification algorithm (TargetScan v8.0) to identify its crucial target transcripts. Our reasoning was that as miR-4787-3p is essential for tumoursphere formation (Figs. 1, 2), leading to a negative ß-score in the 3D tumoursphere screen (Table S1), and as miRNAs act by post-transcriptional repression of direct targets, the upregulation of genes in RNA-seq following its inhibition with positive ß-scores in CRISPR screens would denote crucial target transcripts. Consistent with our hypothesis, the ß-score values of potential miR-4787-3p targets were significantly more positive than those of a random gene list, specifically in the 3D screens for both cell lines (Fig. 4c, g). Among these transcripts, we experimentally validated ARHGAP17, FOXO3A, and PDCD4 as significant direct targets of miR-4787-3p in BC tumourigenesis. ARHGAP17 is thought to have a Tumour Suppressive Role in Colon cancer[36] and Cervical cancer[32]. FOXO3A is a known tumour suppressor and has recently been shown to be antagonised by miRNAs in several cancer types, including in BC[37]. PDCD4 inhibits protein translation to suppress tumour progression and is often decreased in BC. Numerous

regulators including non-coding RNAs control PDCD4 expression in BC. PDCD4 loss is responsible for drug resistance in BC. Modulating the microRNA/PDCD4 axis is suggested as a strategy for overcoming chemoresistance in BC[38]. Interestingly our rescue experiments suggest that, in MCF7, miR-4787-3p mainly acts through the inhibition of PDCD4 whereas in HCC1396, miR-4787-3p mainly acted through the regulation of ARHGAP17.

In conclusion, our study underscores the critical role of miR-4787-3p in BC tumourigenesis, particularly in the aggressive basal-like subtype enriched in tumour-initiating cells (TICs). Targeting miR-4787-3p showed significant inhibition of tumoursphere formation, revealing its potential as a therapeutic target in BC. Moreover, our study suggests that elevated miR-4787-3p expression could serve as a prognostic biomarker for poor outcomes in BC patients. The validated targets of miR-4787-3p, such as ARHGAP17, FOXO3A, and PDCD4, shed light on the molecular mechanisms underlying its oncogenic role in BC. Further investigations and clinical studies focusing on miR-4787-3p inhibition and its prognostic utility hold promise for improved therapeutic strategies and prognostic assessments in BC."

## Materials and methods
### Cell culture
BC cell lines MCF7 and HCC1395 were obtained from Genome Damage and Stability Centre (Sussex) Cell Bank. Cell bank lines were authenticated by ECACC using short tandem repeat (STR) profiling. MCF7 cells were grown in DMEM (Sigma) HCC1395 in RPMI-1640 Medium (Sigma) supplemented with 10% FCS, 2 mmol/l l-glutamine, 100 U/ml penicillin, and 100 mg/ml streptomycin. Between thawing and the use in the described experiments, the cells were passaged no more than 5 times. All cell lines were monthly tested for mycoplasma (MycoAlert, Lonza) and were found negative.

Large scale cell culture was performed in Corning Cell Culture Multi flasks, 3-layer format providing 525 cm$^2$.

Zen and 2'OMe modified miRNA inhibitors were purchased from IDT technologies and transfected with RNAi Max (Fisher). A negative control inhibitor was used as recommended by IDT: NC1 Negative control (human); sequence: ucguuaaucggcuauaauacgc.

### 3D cell culture (tumourspheres)
MCF7 and HCC1395 cells were plated in single-cell suspension in ultralow attachment plates (Corning, # CLS3471). Cells were grown in serum-free DMEM/F12 medium (Gibco) supplemented with B27 (1:50, Gibco), 20 ng/mL basic fibroblast grown factor (bFGF, Biolegend), and 20 ng/mL EGF (Sigma).

For the Gecko screen cells were plated at a density of $5 \times 10^6$ cells per ultra-low attached flask (T75). After 7 days of sphere formation spheres were dissociated with Accutase and $5 \times 10^6$ cells re-seeded into the ultra-low attachment flask. Tumoursphere sample pellets were collected after 4 weeks of sphere formation.

For sphere formation assay, BC cells were plated in ultralow attachment plates (24 well) at a density of $1.5 \times 10^3$ for HCC1395 and $1 \times 10^3$ for MCF7 cells/well, and formed spheres with a size larger than 50 μm were counted under the microscope. The percentage of sphere formation efficiency was calculated as a ratio between the number of formed spheres divided by the number of cells seeded, multiplied by 100. For each condition at least 6 wells of a 24 well plate were counted and 3 independent biological replicates performed.

To measure the corresponding 2D growth the CellTiterGlo 2.0 assay was used (Promega) according to the manufacturers instructions. For each condition at least 4 wells of a 24 well plate were assayed and 3 independent biological replicates performed.

### Plasmids and primers
The lentiviral construct used for the GeCKO library was lentiCRISPRv2 (catalogue no. 52961; Addgene) with psPAX2 (catalogue no. 12260; Addgene) and PMD2.G (catalogue no. 12259; Addgene) as VSV-G envelope-expressing plasmids used with lentiviral vectors to produce lentiviruses.

## CRISPR-CAS9 library preparation

We obtained the GeCKO v2 library from Addgene, amplified it by large scale electroporation with Endura competent cells (Lucigen) then amplified a sample by PCR to produce a library for next generation sequencing (NGS) on the Illumina MiSEq platform. The library passed the quality control checks for library representation.

We produced the GeCKO library viruses by transfecting into HEK293FT cells with two lentiviral packaging vectors and harvesting the supernatant 2 days later. The lentivirus virions were titrated in MCF7 and HCC13965 cells using the functional readout of Puromycin resistance. We generated a mutant cell pool from infecting $180 \times 10^6$ cells at a low MOI of 0.3 to ensure only 1 virus per cell. Cells were selected under Puromycin for 10 days to produce a stable mutant pool then a T0 sample harvested from at least $60 \times 10^6$ cells. The screen target population was then split into 2D and 3D culture conditions and cultured for 4 weeks.

## RNA isolation and RT-qPCR assays/Taqman

Total RNA from cultured cells was extracted from TRIZOL Reagent (Sigma) using Direct-Zol RNA MiniPrep kit (Zymo) following the manufacturer's instructions including DNase I treatment. For gene expression, cDNA was synthesised from 1 µg of purified DNase-treated RNA using RevertAid M-MuLV reverse transcriptase and random hexamer primers (Thermo Scientific), according to the manufacturer's protocols. RT-qPCR assays were performed on a StepOne Real-Time PCR System using Fast SYBR Green Master Mix (both from Applied Biosystems).

## RNAseq—quantseq and analysis

Total RNA from cultured cells was extracted from TRIZOL Reagent (Sigma) using Direct-Zol RNA MiniPrep kit (Zymo) following the manufacturer's instructions including DNase I treatment. Quality and quantity of the extracted RNA samples were assessed with a 2100 Bioanalyzer using RNA 6000 Pico Kit (Agilent).

Single-indexed mRNA libraries were prepared from 100 ng of RNA with QuantSeq 3′ mRNA-Seq Library Prep Kit FWD (Lexogen), according to manufacturer's instructions. Quality of libraries was measured using 2100 Bioanalyzer DNA High Sensitivity Kit (Agilent). Sequencing was performed with BGI DNBseq System (BGI) with PE100 reads. QuantSeq 3′ mRNA-Seq Integrated Data Analysis Pipeline on Bluebee® (Lexogen) was used for preliminary quality evaluation of the RNA sequencing data and Differential Expression analysis was performed on the platform using standard settings for the Quantseq 3′ kit (www.lexogen.bluebee.com). Bluebee uses DESeq2 to generate significant gene expression change and we used p-adjusted < 0.05 to select the differentially expressed genes.

## 3′ UTR reporter assay

3′UTR reporter vectors were purchased from Active Motif/Switchgear Genomics, along with the miR-4787 mimic, control mimic, negative control reporters. Reporters and mimics/inhibitors were reverse transfected with Lipofectamine 3000 (Fisher) and luminescence read after 48 h with the LightSwitch™ Luciferase Assay Kit (Active Motif).

3′UTR sequences are available here: https://switchdb.switchgeargenomics.com.

Product codes for the reporters used were as follows: S810530 PDCD4; S810360 PNKD; S880942 FOXO3A; S801327 BVES; S806225 ARHGAP17.

## Bioinformatics

To identify putative targets of miR-4787-3p, we used TargetScan v8.0 by selecting all the identified potential target transcripts without applying any ranking score.

For RNA-seq analysis the QuantSeq 3′ mRNA-Seq Integrated Data Analysis Pipeline on Bluebee® (Lexogen) was used for preliminary quality evaluation of the RNA sequencing data and Differential Expression analysis was performed on the platform using standard settings for the Quantseq 3′ kit (www.lexogen.bluebee.com). Bluebee uses DESeq2 to generate significant gene expression change and we used $p$-adjusted < 0.05 to select the differentially expressed genes.

For miRNA analysis of patients' samples miRNA-seq expression normalised data and clinical data were downloaded from the Xena browser (https://xenabrowser.net). Plots of miRNA expression values were generated in R version 4.1.0 using the ggplot2 package version 3.3.6. The heatmap was generated using the pheatmap package version 1.0.12. Kaplan-Meier curves to show patient overall survival based on miRNA expression data from the TCGA or the METABRIC were produced using the KMplot (https://kmplot.com/analysis/). Pathway enrichment analysis of our RNA-seq data was performed using Enrichr (https://maayanlab.cloud/Enrichr/), whereas the Gene Set Enrichment Analysis (GSEA) was run through the fgsea package version 1.18.0 in R.

## Statistics and reproducibility

Unless otherwise stated at least 3 separate biological replicates were performed and statistical comparisons between groups were assessed with the Mann Whitney Rank Sum test.

Unless otherwise stated Box plots are described as follows: Centre lines show the medians; box limits indicate the 25th and 75th percentiles as determined by R software; whiskers extend 1.5 times the interquartile range from the 25th and 75th percentiles, outliers are represented by dots.

## Reporting summary

Further information on research design is available in the Nature Portfolio Reporting Summary linked to this article.

## Data availability

The RNA-seq and Gecko data were deposited in the NCBI Gene Expression Omnibus and are accessible through GEO SuperSeries accession number GSE243884. The numerical source data behind the graphs can be found in Supplementary Data file 3. Previously generated data-sets analysed here are available from the DIANA-miTED[24], the TCGA and the NCBI GEO databases (GSE100769)[25]. Stem loop miRNA expression counts transformed to log2(RPM + 1) values were obtained from the GDC TCGA BC datasets from the University of California Santa Cruz (UCSC) Xena database (http://xena.ucsc.edu).

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

## Acknowledgements
The authors would like to thank the Medical Research Council (MR/S000410/1) and Action Against Cancer (AAC); Julian and Cat O'Dell; The Searle Memorial (2005) Charitable Trust; Rebecca Maitland's Triple Negative Breast Cancer Fund; The Bennos Boobs Foundation; and in loving memory of Annie Crouch for funding this study. This work used the computing resources of the UK MEDical BIOinformatics partnership—aggregation, integration, visualisation, and analysis of large, complex data (UK MED-BIO) —which is supported by the Medical Research Council.

## Author contributions
L. C. ideated and designed the study, performed the bioinformatic analysis, supervised research, provided the necessary material and wrote the manuscript. T. S. designed and performed the CRISPR screens and most of the experiments associated with the study and revised the manuscript. S. B. and P. D. performed experiments. J. S. provided material and revised the manuscript. All authors revised and approved the study.

## Competing interests
The authors declare no competing interests
