## [Peer Review File · Communications Biology]

Reviewers' comments:

Reviewer #1 (Remarks to the Author):

In the manuscript titled “CRISPR screens in 3D tumourspheres identified miR-4787-3p as a transcriptional start site miRNA”, Stiff, et al. performed genome-wide CRISPR library screens in two breast cancer cell lines grown in 2D or 3D conditions. They identified several miRNAs essential for cell line-derived tumorsphere formation and proposed such miRNAs, particularly miR-4787-3p, regulate the expression of tumor suppressors. Combining the CRISPR screen hits and transcriptomic analysis, Stiff, et al. demonstrated that miR-4787-3p can target ARHGAP17, FOXO3A and PDCD4 to suppress their expression and therefore execute its oncogenic role in breast cancers. The studies were well designed and the data interpretation is generally consistent with the results presented. However, the following questions need to be addressed before the manuscript can be considered for publication.

Major comments:

1. After the CRISPR screening, several miRNA hits were identified to be critical for tumorsphere growth specifically. Naturally such hits are validated using CRISPR KO in the cell lines. Using miRNA inhibition would be an additional validation approach. In fact, the results presented with inhibitors revealed a much less significant tumorsphere growth suppression as would be expected from the CRISPR KO screens. Such examples include miR-301-3p and miR-1250-5p (Figure 2).
2. Whether miR-4787-3p promotes tumorsphere growth directly by regulating the expression of the potential targets like ARHGAP17, FOXO3A and PDCD4 need to be established. For instance, overexpression or knockdown of individual target or in combination can be tested for tumorsphere growth in the presence or absence of the miRNA or the miRNA with mutations defective in regulating these targets.

Minor comments:

1. In order to specifically examine the miRNA dependency, why can't the authors generate or obtain a focused library targeting miRNA only? It is also suggested that the protein coding genes be analyzed in their roles in the 2D and 3D formats. Would the miR-4787-3p targets be revealed by such analysis?
2. Fig.4, top hits from the RNA sequencing should be validated by quantitative RT-PCR.
3. Several figure legends need revising to have consistent format.

Reviewer #2 (Remarks to the Author):

A) Brief Summary of the Manuscript:

The study elucidated the role of miR-4787-3p in breast cancer (BC) tumorigenesis through innovative methodologies, including pooled CRISPR screens conducted in both 2D and 3D culture models. Subsequent RNA-seq analysis was employed to investigate the expression signatures following the inhibition of miR-4787-3p. The research identified miRNA signatures critical for the formation of BC mammospheres, highlighting the pronounced up-regulation of miR-4787-3p, particularly in basal-like BCs. Contrary to prior assumptions, the study unveiled that miR-4787-3p, situated in the 5'UTR of the protein-coding gene DOCK3, exhibits dependence on both DROSHA and DICER1 for its maturation. Inhibition of miR-4787-3p was found to reduce mammosphere formation, suggesting its potential as a therapeutic target in BC. Moreover, the study proposes elevated miR-4787-3p expression as a promising prognostic biomarker for adverse outcomes in BC patients. The investigation delves further into the molecular mechanisms underlying miR-4787-3p oncogenic role in BC. Experimentally validated gene targets, ARHGAP17, FOXO3A, and PDCD4, underscore the significance of miR-4787-3p in BC.

B) Overall Impression of the Work:

The study stands out for its methodological innovation, employing CRISPR-Cas9 pooled screening targeting miRNAs in both mammosphere and monolayer culture conditions, providing a unique perspective on miRNA function. The integration of experimental findings with comprehensive bioinformatic analyses, including exploration within TCGA and METABRIC databases, adds depth and robustness to the results. Notably, the study challenges previous assumptions regarding the processing of miRNAs located in the 5'UTR of protein-coding genes, showcasing the importance of canonical processing for miR-4787-3p. The authors effectively link their findings to clinical relevance by proposing miR-4787-3p as a potential prognostic biomarker, opening avenues for tailored therapeutic interventions in BC. Overall, the work is characterized by its novelty, methodological rigor, and impactful translational implications.

C) Specific comments, with recommendations for addressing each comment

I. Minor Comments:

- 1) The phrase: "Consequently, crucial miRNAs that possess the ability to impact cancer vulnerabilities might be overlooked, especially when their expression levels remain unchanged under such conditions" may benefit from further clarification or a citation exploring alternative methods to capture differences when expression levels remain unchanged.
- 2) Telomere maintenance is presented in Figure S1.e, not in Figure S1.f as mentioned in the text.
- 3) The biological reproducibility of Figures 2a-b and 2c-d should be clarified. Specify if the calculation is for one 24-well plate, include information about biological replicates, and indicate how many cultures were used for the luminescent read.
- 4) Figure 2.e is not described in the text. Consider adding pictures of basal-like cell lines for a broader perspective on morphological differences in mammospheres.
- 5) In Figure 3.i, the color code for the PAM50 heatmap is confusing. Consider using a different color code to make differences, particularly related to basal & Luminal B, more evident.
- 6) Provide clarification on how the differential gene expression was conducted in the "RNA-seq analysis on cells treated with miRNA inhibitors..." section. Specify whether each miRNA

perturbation was compared against the scramble control of each parental cell line and confirm if the scramble control was the same for both cell lines.

7) Clarify the rationale behind the selection of hsa-miR-4787-3p, as there is also limited evidence associating hsa-miR-1468-5p and miR-6501-5p with BC. Provide details on why this miRNA was chosen, considering the available literature. For instance, in terms of documented functions for miRNAs in your final subset, there are the following references: Liu et al., 2021. MicroRNA-301a-3p promotes triple-negative breast cancer progression through downregulating MEOX2 & Kim et al., 2016. Transcriptome-wide analysis of compression-induced microRNA expression alteration in breast cancer for mining therapeutic targets.

8) Indicate how many cultures were used for the quantification of Figure S5c-d.

9) For the TargetScan prediction, include information about the ranking score method used (version), enhancing transparency and aiding future research criteria selection.

10) In Figure 4, the highlighted red dots are not clear. Consider highlighting genes evaluated downstream.

11) For figures compiling results from MCF7 and HCC3519, consider adding a colored frame to visually identify each cell line for improved aesthetics.

12) BVES is not highlighted in Figure 4.h.

13) The luciferase activity assay and the 3'UTR sequence used for the assay are not described in the methods.

14) Explain why the transfection was performed in MCF7, while two interactions were described in HCC1395. Consider evaluating interactions in HCC1395 for consistency.

15) Standardize the use of "tumor" or "tumour" throughout the document for consistency.

II. Major Comments:

1) Clarify the concept of Tumor Initiating Cells (TICs) and define the criteria used to maintain this assumption, especially in the context of mammosphere culture supplemented with B27. Consider discussing characteristics of mammospheres that reflect TICs or redefine the concept. For instance, take a look at the definition proposed by Wang et al., 2014. Comparison of mammosphere formation from breast cancer cell lines and primary breast tumors.

2) The methods section requires more descriptive documentation of the bioinformatic analysis, including the bioinformatic pipeline used for RNA-seq data analysis, versions of software, and statistical methods employed for significance determination. Add a dedicated method section for this information.

RE: COMMSBIO-23-4771

“CRISPR screens in 3D tumourspheres identified miR-4787-3p as a transcriptional start site miRNA essential for breast tumour-initiating cell growth”.

In answer to the specific comments from each reviewer, we have provided below a point-by-point response (our responses are written in blue):

Reviewer #1:

Major comments:

1. The studies were well designed and the data interpretation is generally consistent with the results presented.

We are grateful to the reviewer for these positive remarks.

2. After the CRISPR screening, several miRNA hits were identified to be critical for tumorsphere growth specifically. Naturally such hits are validated using CRISPR KO in the cell lines. Using miRNA inhibition would be an additional validation approach. In fact, the results presented with inhibitors revealed a much less significant tumorsphere growth suppression as would be expected from the CRISPR KO screens. Such examples include miR-301-3p and miR-1250-5p (Figure 2).

We thank the reviewer for pointing this out and we completely agree with the reviewer that inhibition of miRNA may not be as effective as a full CRISPR knockout, and indeed, this is what we observed.

We opted to validate the miRNA hits using miRNA inhibitors rather than repeating CRISPR KO for several reasons:

1. By using miRNA inhibitors, we aimed to confirm that the effects observed in the screen post miRNA KO were specifically attributable to miRNA depletion rather than off-target effects, providing a complementary approach to validate our findings.

2. CRISPR editing can introduce genomic modifications beyond miRNA repression, potentially altering regulatory regions. Our choice of miRNA inhibition allowed us to focus only on reducing miRNA expression without modifying the genome, ensuring that observed effects were indeed due to the miRNAs rather than to alternative disruption of important regulatory regions.

3. MiRNA inhibition represents a potential cancer therapeutic strategy. Therefore, we think that demonstrating that phenotypic changes still occur following miRNA inhibition is important for assessing its therapeutic relevance and translational potential.

2. Whether miR-4787-3p promotes tumorsphere growth directly by regulating the expression of the potential targets like ARHGAP17, FOXO3A and PDCD4 need to be established. For instance, overexpression or knockdown of individual target or in combination can be tested for tumorsphere growth in the presence or absence of the miRNA or the miRNA with mutations defective in regulating these targets.

This is an excellent idea, and we greatly appreciate this suggestion, which has improved our study. Therefore, we have performed these experiments to address these points:

We first overexpressed the identified targets of miR-4787-3p: ARHGAP17, FOXO3A, and PDCD4, in both MCF7 and HCC1395 cell lines. Indeed, we observed a significant reduction in 3D tumoursphere formation upon overexpression of each gene in both cell lines, validating our model. These new experiments have been included in Figure 6b-c of the revised version of the manuscript.

Furthermore, as suggested by the reviewer, we also performed knockdown experiments targeting ARHGAP17, FOXO3A, and PDCD4 in combination with miR-4787-3p inhibition (4787i) in both cell lines. This allowed us to evaluate the importance of these targets in mediating the 3D tumour formation phenotype induced by miR-4787-3p.

Remarkably, inhibition of miR-4787-3p in combination with the repression of these targets significantly rescues the 3D sphere phenotype, particularly when all three targets were co-depleted along with miR-4787-3p inhibition. This indicates that miR-4787-3p induces tumoursphere formation through the regulation of these targets (new Figure 6d-e of the main manuscript). Interestingly, in MCF7, miR-4787-3p mainly acts through the inhibition of PDCD4, however, when all three targets are inhibited, the sphere formation defect is completely rescued, because it is no longer significant compared to the non-targeting control (NC1i) (new Figure 6d of the main manuscript).

In contrast, in the triple-negative breast cancer cell lines HCC1396, miR-4787-3p mainly acted through the regulation of ARHGAP17. However, only when all three targets were co-inhibited, the sphere formation defect was partially rescued compared to inhibition of miR-4787-3p alone (new Fig 6e of the main manuscript).

Minor comments:

1. In order to specifically examine the miRNA dependency, why can't the authors generate or obtain a focused library targeting miRNA only? It is also suggested that the protein coding genes be analyzed in their roles in the 2D and 3D formats. Would the miR-4787-3p targets be revealed by such analysis?

We evaluated the possibility of using a miRNA-focused library. However, we opted for a combined library in the study because we wanted to test the hypothesis that using a CRISPR library containing gRNAs targeting both miRNAs and protein-coding genes, such as the GeckOv2 library, could help identify key miRNA targets. Indeed, one of the important conclusions of our study is the identification that protein-coding genes positively selected in the CRISPR screens were enriched with putative targets of the negatively selected miR-4787-3p (Figures 4c and 5c).

2. Fig.4, top hits from the RNA sequencing should be validated by quantitative RT-PCR.

Thank you for this suggestion, which further improved our study. We conducted validation experiments on the top targets of miR-4787-3p from both cell lines. These experiments confirmed the RNA sequencing results, showing an increase in target expression after miR-4787-3p inhibition under 3D growth conditions (new Figures 4e and 5e of the revised version of the manuscript). In all but one case, this increase is significant.

3. Several figure legends need revising to have consistent format.

We have revised the legends so that they now have a consistent format throughout.

Reviewer #2:

Overall Impression of the Work:

The study stands out for its methodological innovation, employing CRISPR-Cas9 pooled screening targeting miRNAs in both mammosphere and monolayer culture conditions, providing a unique perspective on miRNA function.

The authors effectively link their findings to clinical relevance by proposing miR-4787-3p as a potential prognostic biomarker, opening avenues for tailored therapeutic interventions in BC. Overall, the work is characterized by its novelty, methodological rigor, and impactful translational implications.

We thank the reviewer for these outstanding remarks about the quality of our manuscript.

I. Minor Comments:

1) The phrase: "Consequently, crucial miRNAs that possess the ability to impact cancer vulnerabilities might be overlooked, especially when their expression levels remain unchanged under such conditions" may benefit from further clarification or a citation exploring alternative methods to capture differences when expression levels remain unchanged.

We apologize for the lack of clarity in this sentence. We have replaced it with the following sentence to enhance clarity, highlighted in red in the revised version of the manuscript:

"For example, a subset of miRNAs might have the ability to impact cancer vulnerabilities, but their expression remains unchanged between cancer and normal samples. In such cases, experimental conditions used to discover important miRNAs are based solely on differences in expression levels, such as RT-qPCR, microarray or RNA-seq might overlook important miRNAs".

2) Telomere maintenance is presented in Figure S1.e, not in Figure S1.f as mentioned in the text.

We have corrected this error in the manuscript.

3) The biological reproducibility of Figures 2a-b and 2c-d should be clarified. Specify if the calculation is for one 24-well plate, include information about biological replicates, and indicate how many cultures were used for the luminescent read.

We have updated the figure legend and methods so that the biological reproducibility is clear.

4) Figure 2.e is not described in the text. Consider adding pictures of basal-like cell lines for a broader perspective on morphological differences in mammospheres.

We have amended the manuscript to mention Fig 2e. We state in the text that HCC1395 tumourspheres had very similar gross morphology to MCF7 tumourspheres.

5) In Figure 3.i, the color code for the PAM50 heatmap is confusing. Consider using a different color code to make differences, particularly related to basal & Luminal B, more evident.

We have changed heatmap colour scheme to improve clarity.

6) Provide clarification on how the differential gene expression was conducted in the "RNA-seq analysis on cells treated with miRNA inhibitors..." section. Specify whether each miRNA perturbation was compared against the scramble control of each parental cell line and confirm if the scramble control was the same for both cell lines.

We have updated this part of the Results section for clarity and added more detail to the Methods. We have corrected the manuscript. Each miRNA perturbation was compared against a scramble negative control of each parental cell line. The scramble control was the same in both cell line (NC1 from IDT DNA).

NC1 Negative control (human) `ucguuaaucggcuauaaucgc`

7) Clarify the rationale behind the selection of hsa-miR-4787-3p, as there is also limited evidence associating hsa-miR-1468-5p and miR-6501-5p with BC. Provide details on why this miRNA was chosen, considering the available literature. For instance, in terms of documented functions for miRNAs in your final subset, there are the following references: Liu et al., 2021. MicroRNA-301a-3p promotes triple-negative breast cancer progression through downregulating MEOX2 & Kim et al., 2016. Transcriptome-wide analysis of compression-induced microRNA expression alteration in breast cancer for mining therapeutic targets.

Thank you to the reviewer for this suggestion. We now realise that the rationale behind this choice was indeed not very clear. We selected miR-4787-3p for further analysis for two reasons: 1) It has never been described in breast cancer before. 2) It belongs to the class of transcriptional start site miRNAs, which have been previously described but never associated with cancer.

We have clarified this in the text of the revised version of the manuscript and highlighted the changes in red. To provide additional information about other miRNAs, we included the articles suggested by the reviewer in the discussion: Liu et al., 2021 ("MicroRNA-301a-3p promotes triple-negative breast cancer progression through downregulating MEOX2") and Kim et al., 2016 ("Transcriptome-wide analysis of compression-induced microRNA expression alteration in breast cancer for mining therapeutic targets"). These articles document the description of miR-301a-3p and miR-4446-3p in breast cancer.

We have also included that Yerukala Sathipati et al., 2018 ("Identifying a miRNA signature for predicting the stage of breast cancer") identified that miR-1468 is part of a miRNA signature that predict the stages of breast cancer. Additionally in the discussion we say "miR-4446 is part of a miRNA signature predicting high-risk BC cases from liquid biopsies. Farina et al., 2017: ("Development of a predictive miRNA signature for breast cancer risk among high-risk women"), while miR-6501 serves as a diagnostic and prognostic indicator in breast cancer: Ramanto et al., 2021

("The regulation of microRNA in each of cancer stage from two different ethnicities as potential biomarker for breast cancer").

8) Indicate how many cultures were used for the quantification of Figure S5c-d.

We have updated the legend to include this information.

9) For the TargetScan prediction, include information about the ranking score method used (version), enhancing transparency and aiding future research criteria selection.

We have added this information to the Methods section.

10) In Figure 4, the highlighted red dots are not clear. Consider highlighting genes evaluated downstream.

As suggested in Fig 4b and 5b we have highlighted genes of interest in cancer (green), those validated by qPCR (orange) and those followed up for further evaluated downstream (red).

11) For figures compiling results from MCF7 and HCC3519, consider adding a colored frame to visually identify each cell line for improved aesthetics.

As suggested, we have adjusted the figures to improve clarity. MCF7 text is coloured red and HCC1395 green. We have split figure 4 into two figures to better distinguish MCF7 and HCC1395 experiments.

12) BVES is not highlighted in Figure 4h.

We have highlighted BVES in the figure.

13) The luciferase activity assay and the 3'UTR sequence used for the assay are not described in the methods.

We have added this to the Methods section and included links to the 3'UTR sequences used in the reporter vectors.

14) Explain why the transfection was performed in MCF7, while two interactions were described in HCC1395. Consider evaluating interactions in HCC1395 for consistency.

We have now validated miRNA regulation of their targets in both cell lines by inhibition miR-4787-3p following measure of the mRNA targets by RT-qPCR. These results are now in Figure 4e and 5e.

Please, see also response to reviewer #1.

15) Standardize the use of "tumor" or "tumour" throughout the document for consistency.

We have amended the manuscript to "tumour" throughout.

II. Major Comments:

1) Clarify the concept of Tumor Initiating Cells (TICs) and define the criteria used to maintain this assumption, especially in the context of mammosphere culture supplemented with B27. Consider discussing characteristics of mammospheres that reflect TICs or redefine the concept. For instance, take a look at the definition proposed by Wang et al., 2014. Comparison of mammosphere formation from breast cancer cell lines and primary breast tumors.

We thank the reviewer for this suggestion which increased the quality of our manuscript. We have now improved this aspect by including the characteristics of mammospheres as defined by Wang et al., as the reviewer suggested.

To this end, we have included in the introduction of the revised version of our manuscript the following:

CSCs are also called TICs, because when inoculated into severe combined immunodeficiency disease (SCID) mice, represent the minority of breast cancer cells capable of forming new tumours and are CD44⁺/CD24^{-/low}/lineage⁻. TICs can be isolated or enriched through various methods. This includes sorting breast cancer cells based on the CD44⁺/CD24^{-/low} phenotype, or culturing cells under non-adherent, non-differentiating conditions to promote the formation of tumourspheres. Interestingly, studies have demonstrated that adding B27 to the culture media in non-adherent conditions improves tumoursphere formation efficiency and enhances the enrichment of the CD44⁺/CD24^{-/low} lineage, which can reach up to approximately 95% in breast cancer cell lines.

2) The methods section requires more descriptive documentation of the bioinformatic analysis, including the bioinformatic pipeline used for RNA-seq data analysis, versions of software, and statistical methods employed for significance determination. Add a dedicated method section for this information.

We have added a Bioinformatics section to the Methods as suggested and included more detail.

Reviewers' comments:

Reviewer #1 (Remarks to the Author):

My questions have been adequately addressed in the revised manuscript. Just one last minor point: overexpression and knockdown evidence such as Western blot should be provided to substantiate the data in Figure 6.

Reviewer #2 (Remarks to the Author):

I have reviewed the revised manuscript and I am pleased to report that all of my previous comments have been appropriately addressed and fulfilled by the authors. The revisions have significantly improved the clarity and quality of the manuscript.

Regarding the minor misspellings mentioned, I noticed that the words "utilised", "normalised", and "antagonised" are spelled incorrectly in several instances throughout the manuscript. I recommend correcting these minor errors.

Overall, I am satisfied with the responses provided by the authors. I have no further comments to add at this time.

REVIEWERS' COMMENTS:

Reviewer #1 (Remarks to the Author):

I have no additional questions